# Evaluation Instruments for Assessing Back Pain in Athletes: A Systematic Review Protocol

**DOI:** 10.3390/healthcare8040574

**Published:** 2020-12-18

**Authors:** Vinicius Diniz Azevedo, Regina Márcia Ferreira Silva, Silvia Cristina de Carvalho Borges, Michele da Silva Valadão Fernandes, Vicente Miñana-Signes, Manuel Monfort-Pañego, Priscilla Rayanne E Silva Noll, Matias Noll

**Affiliations:** 1Faculdade de Educação Física e Dança, Universidade Federal de Goiás, Samambaia Campus, Goiânia 74690-900, Brazil; silviaborges@discente.ufg.br; 2Department of Public Health, Instituto Federal Goiano, Ceres 76300-000, Brazil; regina.silva@ifg.edu.br (R.M.F.S.); michele.fernandes@estudante.ifgoiano.edu.br or priscilla.noll@usp.br (P.R.E.S.N.); 3Universitat de València, 46010 València, Spain; vicente.minana@uv.es (V.M.-S.); manuel.monfort@uv.es (M.M.-P.); 4Department of Obstetrics and Gynecology, Faculdade de Medicina FMUSP Universidade de São Paulo, São Paulo 05403-000, Brazil

**Keywords:** athletes, sportsmen, back pain, instrument, questionnaire

## Abstract

Back pain is a public health problem that affects adolescents and adults worldwide. However, studies on back pain present inconsistent findings in part due to the use of different instruments, especially for athletes. Therefore, the objective of this systematic review protocol was to map the existing evidence on such tools. The systematic review will be conducted according to PRISMA guidelines. Five electronic databases, Embase, MEDLINE, SPORTDiscus, CINAHL, and Scopus will be searched. This review includes studies that investigated prevalence, incidence, and other variables. Titles and abstracts will be selected. Two independent reviewers will read the articles carefully and discrepancies, if any, will be dealt with by a third reviewer. All steps will be completed with Rayyan for systematic reviews and the methodological quality will be analyzed with a COSMIN checklist. Discussion: This systematic review will gather evidence on tools that assess back pain in athletes. The findings may indicate the most appropriate tools for assessing back pain. They will contribute to better reliability, safe measurements, and help to standardize a comparison tool between different studies. They will also assist in the development of specific tools for athletes. Registration: This review was submitted and registered under CRD42020201299 in PROSPERO.

## 1. Introduction

Pain is defined by the International Association for the Study of Pain as an “unpleasant sensory and emotional experience associated with actual or potential tissue damage, or described in terms of such damage” [1,2]. Pain can affect several body parts, and when it affects the spine, it can cause malaise, discomfort, and even postural changes [3,4]. Recent studies have associated discomfort due to pain with several factors [5] such as age [6,7,8], sex [6,9], sexual activity [10], sleep [11], and body mass index (BMI) [12]. Additionally, back pain affects the formation of structures in the lumbar region [13,14] and causes knee [15] and shoulder [16] pain during high impact sports [17,18]. Thus, it has been studied for several sports such as rowing, rugby, football, fighting, and gymnastics [19,20,21,22,23,24]. In the literature, systematic reviews have concluded that age, advanced and late puberty, family history of back pain, asthma, headaches, abdominal pain, depression, and anxiety are potential risk factors that trigger back pain [25,26].

In addition, back pain poses a significant economic burden on public health, thus requiring attention. Spanish researchers concluded that the costs attributable to back pain were 8945.6 million euros, of which 74.5% were involved indirect costs, representing 0.68% of the total Spanish gross domestic product [27]. The problem of the heterogeneity of data sources in the analysis of the prevalence and incidence of low back pain (LBP) has been pointed out in other review studies [28].

There is evidence that heterogeneity of data also occurs in studies on specific populations of athletes, thus highlighting the need to unify criteria contributing to the development of valid instruments. On the contrary, as previously mentioned, LBP has a significant effect on the economy and quality of life of citizens [29]. Thus, it is essential to create a global database that collects large-scale information on back health, considering the specific lifestyle habits of the population, to be able to deepen the study of its large-scale effect. An increasing number of questionnaires are also available for use in research and in the assessment of back pain in different populations [30,31,32].

Some of the tools recommended for non-specific pain are the Roland–Morris Disability Questionnaire [33] and the Numeric Rating Scale for Pain Intensity Assessment [34]. Examples of others tools are the Oswestry Disability Index (ODI) [35], EuroQoL [36], the Standardized Nordic Questionnaire [37], the Back Pain and Body Posture Evaluation Instrument (BackPEI) [38], and the Young Spine Questionnaire [39]. Among other questionnaires, the Japanese Orthopedic Association’s Back Pain Assessment Questionnaire (JOABPEQ) [40] is a simple and effective tool for evaluating back pain. The intensity and incapacity level due to back pain can also be gauged using the Chronic Pain Grade Questionnaire (CPGS), which is suitable for all chronic pain conditions including chronic musculoskeletal (MSK) and back pain [41].

In recent years, several tools for assessing back pain in athletes have been developed or adapted [20]. However, the definition of “athlete” varies [42,43,44], which can influence the criteria for selecting participants for studies. In general, the term “athlete” is related to the ability to perform tasks or activities such as sports [44,45]. In order to facilitate standardization and precision in the definition, researchers have categorized individuals based on the scores they obtained in the following: exercise intent, exercise volume (hours/week), level of competition associations between exercises [44], and regularly participation in sports competitions [44,46,47].

Why is it important to have a specific instrument that assesses back pain in athletes? Several instruments have been developed for the normal population; however, athletes are subject to more intense and sport-specific variables that are not included in these instruments. For instance, variables that may be risk factors for back pain (such as volume of exercise, level of competition, unilateral practices, overuse) and how back pain affects sports practice and sport competition as well as possibly leading to fear-avoidance of exercise are essential in this scenario. Therefore, the development of specific instruments to evaluate back pain in athletes is necessary. A tool often used to assess back pain and functional abilities in athletes is the Micheli Functional Scale (MFS) [48], whereas the Athletes Disability Index [10] is employed to evaluate disability due to LBP. However, using different tools, instead of a standardized model for pain assessment, can lead to divergent findings [10]. The heterogeneity of assessment tools in clinical trials also prevents comparisons among studies and systematic reviews [49].

Health professionals (such as doctors, physiotherapists, nurses, and physical education professionals) around the world increasingly need reliable and valid back pain assessment tools at their disposal. They intend to measure, monitor, assess, and diagnose back pain using specific validated tools. Accordingly, certain reviews have already sought to study back pain assessment tools [50,51]. However, no review has been conducted on pain assessment tools for athletes. Therefore, this systematic review aimed to identify tools that assess back pain in athletes. We intend to systematically review each tool by examining and comparing its features and limitations. The primary result may confirm the existing instruments that evaluate back pain in athletes and inform on their specificities and appropriateness, thus contributing to a more reliable, safe, and standardized measurement that allows study comparisons. The secondary results could contribute to the development of specific assessment tools for athletes.

Therefore, before developing a new test or measure, it is necessary to identify the existing instruments that measure the construct of interest, which will allow us to identify the specific instruments in use for the desired proposal. This will also help verify whether, for example, the reliability and validity of the instruments are well established. Likewise, it is possible to examine whether the test was assessed using reliability estimates (for example, internal consistency and test-retest) and varied strategies for establishing validity (for example, concurrent content and validity) as well as more extensive evidence of construct validity in varied populations [52].

## 2. Materials and Methods

This protocol was designed based on items from the Preferred Reporting Items for Systematic Reviews and Meta-analyses (PRISMA) [53]. This review was submitted and registered under CRD42020201299 in the International Register of Prospective Systematic Reviews (PROSPERO). This register aims to increase transparency and reproducibility while avoiding duplicate efforts on the same topic.

### 2.1. Identification of the Search Question

The main question of the study is: What are the existing tools, published in literature, that assess back pain in athletes?

### 2.2. Eligibility

#### 2.2.1. Inclusion Criteria

Articles which will be included in the review, irrespective of the language and publication date, must meet the following criteria: (a) questionnaires that mentioned prevalence, incidence, intensity, location, functional disability, or other variables related to back pain; (b) questionnaires that evaluated athletes and variables related training and sports; and (c) the assessment tools can be created, adapted, or translated, but must be definitely validated or at least tested for their reproducibility.

In this context, “back pain” will be defined as “pain in the cervical, thoracic, and/or lumbar areas” [25,26,54,55]. Moreover, “athlete” will be defined as individuals who participate in sports competitions and are engaged in training activities for four or more hours per week (volume of exercise) [44,47,56].

#### 2.2.2. Exclusion Criteria

Articles will be excluded if they are (a) systematic reviews, reports, case studies, or opinion articles; (b) studies that include individuals with physical or mental disabilities; and (c) studies conducted on individuals with chronic diseases, pregnant women, lactating women, and people with spinal fractures or recent surgeries.

### 2.3. Identification of Relevant Studies

The search terms and synonyms were separately selected from three main categories and finally combined into one search sequence per database. Therefore, the three main descriptors included “athletes”, “instrument”, and “back pain”. Appendix A (Table A1) presents the logical structure of the general search strategy with all the descriptors and Boolean operators that will be used in all three databases.

### 2.4. Data Management

The results of the search strategies will be imported into the Mendeley software, where duplicate articles will be identified and removed. The first selection stage will be based on the title and abstract, wherein each article will be evaluated as per the eligibility criteria. After this stage, the remaining articles will be read entirely to confirm their eligibility. All steps will be performed using the Rayyan software, which aids in rapidly surveying and filtering studies eligible for systematic reviews [57].

### 2.5. Study Selection

All steps will be conducted independently by two reviewers (V.D.A and R.M.F.S.), and disagreements, if any, will be resolved by a third reviewer (M.N.). The reliability among evaluators for the classifications of individual components will be determined by calculating the percentage of agreement and Cohen’s Kappa coefficient. Finally, eligible articles will be included in the systematic review. The flowchart for this systematic review is shown in Figure 1.

### 2.6. Training of Reviewers

Researchers who will participate in eligibility assessments will be trained on the study inclusion/exclusion criteria and will practice assessing the eligibility of 50 abstract samples before they start to code the articles [55]. They will also be instructed to use tools for assessing the risk of bias in five excluded articles. Furthermore, they will conduct standardized analyses with the Mendeley and Rayyan software.

### 2.7. Data Extraction

The following data will be extracted from the selected studies: characteristics of the sample, namely title, author, place, study groups, average age, gender proportions, follow-up, objective of the study, year of publication, and the sport investigated; details on didactic intervention programs, such as assessment methodology and instruments; specific baseline and follow-up outcome data, and the definitions of athlete and back pain. Our systematic review focused on the analysis of published articles (secondary data) and did not require ethical approval. Upon completion, the authors will submit the systematic review for publication in a peer-reviewed journal.

### 2.8. Methodological Quality and Data Synthesis

The methodological quality of the articles will be evaluated using the Consensus-based Standards for the Selection of Health Measurement Instruments (COSMIN). This checklist aims at facilitating the selection of result measures reported by study participants. It also evaluates the measurement properties of the health assessment tools [58]. The COSMIN checklist consists of 114 items, divided into twelve criteria. Each of the 114 items is scored on a 4-point rating scale (excellent, good, fair, and poor). Within each of the twelve criteria, the item with the lowest rating is considered [59,60].

Using COSMIN, a ten-step procedure is developed to perform a systematic review, specifically to obtain measures for results reported by the patient. Steps 1–4 refer to the preparation and execution of bibliographic research and selection of relevant studies. Steps 5–8 are concerned with evaluating the quality of eligible studies, measurement properties, and interpretability and feasibility aspects. Steps 9 and 10 refer to formulating recommendations and reporting a systematic review [61].

For steps 5–8, the methodological quality will be independently evaluated by two reviewers (V.D.A and R.M.F.S.), and disagreements, if any, will be resolved by a third reviewer (M.N.). Therefore, to evaluate the methodological quality of the studies included using the COSMIN Risk of Bias checklist, one must first determine the measurement properties to be evaluated in each article. The quality of each study is evaluated separately using the corresponding COSMIN box. Thus, each study is evaluated as very good, adequate, doubtful, or inadequate quality. To determine the overall quality rating of each study, the lowest rating of any standard in the box is taken [61].

We will present an overview of the available studies, and all the included studies will be evaluated for their quality. A summary of the best available evidence for each measurement property will then be summarized. This article will select a preliminary set of result-measuring instruments for the central domains, chosen from those often used for evaluating back pain in athletes. In addition, we will analyze those recommended by other initiatives that aim to standardize measurements for LBP or chronic pain.

## 3. Results

This systematic review protocol will seek publications on tools that assess back pain in athletes. However, some limitations to this research need to be highlighted. The heterogeneity of questionnaires can lead to different methods of questionnaire elaboration. Many of them may not have clearly defined the term “athlete”, and this may cause unequal sample selection (such as age and hours of training). Finally, some articles may present outdated instruments. Therefore, systematic reviews of the result measuring instruments are important tools to select the most suitable instrument to measure something desired by a specific study population. Consequently, they should be selected carefully.

## 4. Conclusions

Aside from presenting reliable and valid tools, this work will allow health professionals such as doctors, physiotherapists, nurses, and physical education professionals to better understand the symptoms of back pain in athletes and respond appropriately. In addition, this work may support programs and actions to provide back pain in this specific group. Therefore, lack of a valid review on back pain assessment tools (especially for athletes) may make this impossible or affect the conduct of research and publications. Finally, we believe that this review can help in the development of specific instruments to assess back pain among athletes.

## Figures and Tables

**Figure 1 healthcare-08-00574-f001:**
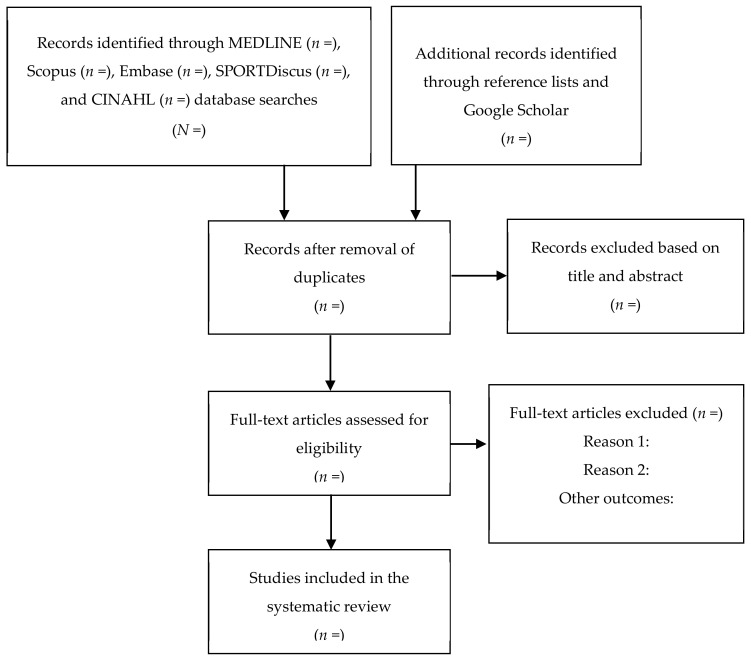
Study selection for systematic review.

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
