# Peer review of "Evaluation Instruments for Assessing Back Pain in Athletes: A Systematic Review Protocol"

_healthcare, 2020, doi:10.3390/healthcare8040574_

Round 1

Reviewer 1 Report

General comments

This manuscript attempts to critically evaluate evaluation instruments for back pain specifically in the population of athletes. In general, the manuscript is concise and well written.

Specific comments

  1. Introduction

- Page 2, Line 47: What does “this” mean?

- Page 2, Line 59: The ODI is already mentioned in Line 57.

- Page 2, Line 72: What is “them”? Avoid using vague pronouns.

- Page 2, Line 72-74: I agree with the authors that evaluation instruments that assess functional disability for the general population may not be suitable for athletes. However, the argument really depends on what aspect of back pain is assessed. I don’t think that there is much difference for evaluation instruments, such as the Numeric Rating Scale for Pain Intensity Assessment.

  1. Materials and Methods

- Page 3, Line 110: The authors mentioned that the definition of an athlete varies in the literature. What is the operational definition of “athletes” that will be used when assessing eligibility of the articles? Clinical presentation of back pain also varies greatly, for example localized pain in the lumbar region or radiating pain to the lower extremity. Please provide more detail on the definition “back pain”. The lack of a clear definition can influence comparisons between studies.

- Page 3, Line 122: “The table below presents…”. Missing table?

- Page 5, Line 161-165: There is no description about synthesis of results. How will the results of studies be combined? How will the heterogeneity between studies be evaluated?

Author Response

General comments

This manuscript attempts to critically evaluate evaluation instruments for back pain specifically in the population of athletes. In general, the manuscript is concise and well written.

Authors: Thank you so much for your time and feedback.

  1. Introduction.

Page 2, line 47: What does "this" mean?

Authors: Thanks for your careful review. The term "this" refers to the problem of heterogeneity of data sources. However, we reorganized the information in the sentence. The change is present, as follow:                       

Page 02, Line 47: “There is evidence that heterogeneity of data also occurs in studies on specific populations of athletes …”

Page 2, Line 59: ODI is already mentioned in line 57.

Authors: Thanks for your suggestion. Based on it, we have removed the reference in the line 57.

Page 2, Line 72: What are "them"? Avoid using vague pronouns.

Authors: The term "them" refers to questionnaires. Therefore, we reorganized the information in the sentence, as follow:

Lines 70-71: “Several instruments have been developed for...”

 Page 2, Line 72-74: I agree with the authors that evaluation instruments that assess functional disability for the general population may not be suitable for athletes. However, the argument really depends on what aspect of back pain is assessed. I don't think there is much difference for the assessment instruments, such as the Numerical Rating Scale for Assessing Pain Intensity.

Authors: Thank you so much for you point. We really agree with the reviewer and we have changed this sentence as follow:

Page 02, Line 70-75: “Several instruments have been developed for the normal population; however, athletes are subject to more intense and sport-specific variables that are not included in these instruments. For instance, variables that may be risk factors for back pain (such as volume of exercise, level of competition, unilateral practices, overuse) and how back pain affects sports practice and sport competition as well as can lead to fear-avoidance of exercise are essential in this scenario.”

  1. Materials and methods

- Page 3, Line 110: The authors mentioned that the definition of an athlete varies in the literature. What is the operational definition of “athletes” that will be used when assessing eligibility of the articles? Clinical presentation of back pain also varies greatly, for example localized pain in the lumbar region or radiating pain to the lower extremity. Please provide more detail on the definition “back pain”. The lack of a clear definition can influence comparisons between studies.

Authors: Thank you so much for you concern. We agree with the reviewer and we have included the operational definition for “athlete” and “back pain” We have improved our inclusion criteria section, as follow:

Page 03, lines 117-120: “In this context, “back pain” will be defined as "pain in the cervical, thoracic and/or lumbar areas"[25, 26, 54, 55]. Moreover, “athlete” will be defined as individuals who participate in sports competitions and are engaged in training activities for 4 or more hours per week (volume of exercise) [44, 47, 56]”.

Page 3, Line 122: "The table below shows ...". Missing table?

Authors: Thanks for the careful review. The Table 1 is as an Appendix. We have rewritten this sentence as follow:

Page 04, Line 130: “Appendix A (Table 1) presents the logical structure of the general search strategy with all the descriptors and Boolean operators that will be used in all the three databases”.

Page 5, Line 161-165: No description of the results overview. How will the results of the studies be combined? How will heterogeneity between studies be assessed?

Authors: We agree that our manuscript was missing this important part. We will provide an overview of available studies and all studies included will be evaluated for their quality. A summary of the best existing evidence for each measurement property will then be summarized. This article will select a preliminary set of result measurement tools for the core domains, choosing from those often used to evaluate athletes' back pain. In addition, we will review those recommended by other initiatives aimed at standardizing measurements for back pain, or chronic pain.

We have included more clearly this statement in the end of the “2.8. Methodological quality and data synthesis” section.

Reviewer 2 Report

Firstly, with regard to questions of form, we must say that it is a paper that complies with the norms established by this journal, being characteristic of this work its pertinent format, design, presentation of the written discourse, structure and organization of the information, showing in a clear way the monitoring of the scientific method, distributing in a balanced and coherent way the information in each one of the sections, from the introduction to the conclusions.

Secondly, regarding content aspects, we must express that this research is totally relevant, since it responds to the current needs that the society demands and that, undoubtedly, the professions related to the Health Sciences require. There is a great deal of research on the development and consolidation of Health Sciences.

Undoubtedly, this paper is perfectly triangulated in the presentation of the contents but I think the results and conclusions were clearer.

Author Response

Comments and Suggestions for Authors:

Firstly, with regard to questions of form, we must say that it is a paper that complies with the norms established by this journal, being characteristic of this work its pertinent format, design, presentation of the written discourse, structure and organization of the information, showing in a clear way the monitoring of the scientific method, distributing in a balanced and coherent way the information in each one of the sections, from the introduction to the conclusions. Secondly, regarding content aspects, we must express that this research is totally relevant, since it responds to the current needs that the society demands and that, undoubtedly, the professions related to the Health Sciences require. There is a great deal of research on the development and consolidation of Health Sciences. Undoubtedly, this paper is perfectly triangulated in the presentation of the contents. I think the results and conclusions were clearer.

Authors: We are very grateful for your review and positive feedback. We are at your disposal for any clarifications.